# Virtual Reality Potentiates Emotion and Task Effects of Alpha/Beta Brain Oscillations

**DOI:** 10.3390/brainsci10080537

**Published:** 2020-08-10

**Authors:** David Schubring, Matthias Kraus, Christopher Stolz, Niklas Weiler, Daniel A. Keim, Harald Schupp

**Affiliations:** 1Department of Psychology, University of Konstanz, 78457 Konstanz, Germany; harald.schupp@uni-konstanz.de; 2Department of Computer and Information Science, University of Konstanz, 78457 Konstanz, Germany; matthias.kraus@uni-konstanz.de (M.K.); niklas.weiler@uni-konstanz.de (N.W.); keim@uni-konstanz.de (D.A.K.); 3Department of Psychology, University of Marburg, 35032 Marburg, Germany; chris.stolz@staff.uni-marburg.de

**Keywords:** alpha oscillations, arousal, attention, emotion, virtual reality

## Abstract

The progress of technology has increased research on neuropsychological emotion and attention with virtual reality (VR). However, direct comparisons between conventional two-dimensional (2D) and VR stimulations are lacking. Thus, the present study compared electroencephalography (EEG) correlates of explicit task and implicit emotional attention between 2D and VR stimulation. Participants (*n* = 16) viewed angry and neutral faces with equal size and distance in both 2D and VR, while they were asked to count one of the two facial expressions. For the main effects of emotion (angry vs. neutral) and task (target vs. nontarget), established event related potentials (ERP), namely the late positive potential (LPP) and the target P300, were replicated. VR stimulation compared to 2D led to overall bigger ERPs but did not interact with emotion or task effects. In the frequency domain, alpha/beta-activity was larger in VR compared to 2D stimulation already in the baseline period. Of note, while alpha/beta event related desynchronization (ERD) for emotion and task conditions were seen in both VR and 2D stimulation, these effects were significantly stronger in VR than in 2D. These results suggest that enhanced immersion with the stimulus materials enabled by VR technology can potentiate induced brain oscillation effects to implicit emotion and explicit task effects.

## 1. Introduction

### 1.1. General Introduction

The advent of better and cheaper virtual reality (VR) consumer hardware also stimulated (neuro) psychological attention and emotion research with more immersive VR environments. In general, both presence, or the feeling of being present in VR, and immersion, or the extent to which a display system can deliver a vivid illusion of VR [1], are strongly correlated with emotion and attention processes (for a review see [2]). Moreover, recent advancements in immersive technology, such as user-tracking, the use of stereoscopic visuals and wider fields of view, also enabled greater perceived immersion [3]. Available evidence suggests that immersion, presence and emotion are related only for arousing, but not for non-arousing neutral content. However, the exact relationship between immersion, presence and emotion remains unclear [2,4], which underlines the importance of research on emotion and attention effects in VR.

When evaluating neuropsychological markers of emotion and attention together with VR, two main disciplines are involved. On the one hand, there is an increasing interest in neuropsychology to use more immersive VR for better affective stimulation (for a review see [5]): Examples include, e.g., to study the impact the context can have on emotional EEG correlates, such as the late positive potential (LPP) [6], assess attentional EEG correlates, such as the P300 in an oddball paradigm for further use in neurorehabilitation [7] or to elicit emotional states, which were then classified by their EEG signal [8]. On the other hand, in the field of human-computer interaction, the question is asked—what can EEG correlates of attention and emotion tell us about the quality of VR? For instance, Kober and Neuper [9] presented irrelevant auditory stimuli while people navigated through a virtual environment. When people felt more present in VR, the ERPs to distracting stimuli were smaller. These findings provide evidence that VR can make valuable contributions to affective neuroscience and vice versa.

While the presentation of fully immersive virtual reality stimuli allows for more complex stimulation, VR stimulation is not always better than conventional presentation on a 2D screen [10]. Virtual reality induced side effects (VRISE) such as motion sickness may persist even with better technology [11] and can be detrimental to the VR experience. Therefore, it should be evaluated which advantages and disadvantages VR stimulation has compared to 2D stimulation if both are potentially viable for a study. However, direct comparisons between VR and 2D are sparse and only partially controlled, e.g., for visual angle. For task related explicit attention, some comparisons exist: An early study found no differences of P300 brain responses for a counting task performed either in VR or 2D [12]. Another pilot study compared P300 brain responses in 2D and VR when participants were asked to spot a partially occluded target within an oddball paradigm [13]. They found a stronger P300 for non-occluded targets in VR, which diminished as occlusion levels rose. Furthermore, the P300 latency became more delayed in VR compared to 2D with higher occlusion levels. Another study compared spatial navigation with a Single-Wall-VR system and a conventional 2D desktop stimulation and found more task related alpha changes in VR [14]. However, the VR stimuli were also presented with a bigger visual angle. Thus, advantages of VR stimulation could be driven by size differences rather than reflecting better immersion. This reasoning is supported by the finding that picture size modulated the magnitudes of cortical responses to emotional stimulus processing [15]. In a similar experiment on spatial navigation in VR and 2D with controlled stimulus size, an increased sense of presence and more EEG theta power was observed for the VR stimulation, but also less postural stability [16]. Other studies comparing EEG correlates of cognitive load in VR and 2D reported mixed findings. While one study found less cognitive load in VR compared to 2D [17], another found more cognitive load and even less learning in VR compared to 2D [18]. Taken together, most studies found increased effects for explicit task attention in VR compared to 2D.

Implicit emotional attention might be a more meaningful metric than explicit task attention because emotional arousal is seen as a crucial element for research in virtual reality, but its role remains still unexplained (for a review, see [2]). It has been shown that emotional responses, such as arousal ratings, are enhanced in VR, e.g., by using either a “low immersion” presentation or a “high immersion” with surrounding presentation [19]. However, neuropsychological markers of emotion, such as the LPP [20] or alpha/beta event related desynchronization or α-ERD [21], have not yet been directly compared between VR and 2D. Especially if the impact of VR is moderated by arousal and emotional stimulus significance, evaluating neuropsychological correlates of emotional processing is critical to assess the impact of VR.

To provide a comparison of virtual reality and conventional 2D stimulation, the paradigm included established manipulations of implicit emotional stimulus significance and explicit attention processes. Specifically, avatars with different emotional facial expressions (angry and neutral) were presented in either VR or conventional 2D environment, controlling the stimuli with regard to size and distance. Furthermore, as explicit task manipulation, participants were asked in different blocks to either count angry or neutral facial expressions. Effects were measured with neuropsychological markers of attention and emotion, including the LPP and P3 as event related potentials and frequency effects from 1–40 Hz, i.e., including power in theta, alpha and beta frequency bands.

### 1.2. Research Questions

(1) Is there a main effect of VR stimulation on ERPs and frequencies, i.e., does it enhance overall responses because of the immersion, or does it diminish responses because of the distraction from unfamiliar stimulation? (2) Does VR stimulation interact with emotion and task manipulations, i.e., does it specifically enhance or attenuate the processing of emotional and target stimuli? Furthermore, questionnaires on immersion and discomfort were used to test behavioral effects.

## 2. Materials and Methods

### 2.1. Participants

Sixteen healthy volunteers (8 males/8 females) with a mean age of 24.8 years (*SD* = 3.7) were recruited on the campus of the University of Konstanz. All participants had normal or corrected-to-normal vision. 7 out of 16 participants had previous experience with virtual reality and within those, exposure to virtual reality in the last month was low (less than 1 h in total). Sample size was determined based on previous research reporting strong and consistent effects on the modulation of event-related potentials and oscillations with samples of sixteen participants (e.g., [21,22]). Participants received monetary compensation or course credit for participation. The ethical committee of the University of Konstanz approved the experimental procedure in accordance with the regulations of the Declaration of Helsinki, and all methods were carried out in full compliance with the approved guidelines (IRB statement 10/2020). All participants provided informed consent.

### 2.2. Stimuli & Setup

Stimulus selection was based on previous research on conventional 2D and virtual reality presentations. Previous studies examining emotional stimulus processing using 2D presentations often focused on the comparison of angry and neutral facial expressions (e.g., [23,24,25]). In addition, stimulus materials were available from a previous VR study [6] in which angry facial expressions have been shown to elicit emotional responses both in a questionnaire and on a neuronal level (i.e., eliciting an LPP). Specifically, the character set consisted of five different characters with three different facial expression configurations each. Figure 1 shows two exemplary characters from the set with the mimic configurations neutral (Figure 1a), angry (Figure 1b) and angry with open mouth (Figure 1c).

The experimental room was furnished with a chair in the center and a small TV cupboard in front of it (see Figure 2a). The study prototype was developed with the game engine Unity3D (https://www.unity3d.com/). In the virtual reality environment, the physical laboratory was reconstructed, including the cupboard and the hand rest of the chair (see Figure 2c). The virtual room was then precisely matched to the real environment to create a so-called “Substitutional Reality Environment” [26], in which each physical object is displayed in the virtual environment as a virtual substitute at the same place and size as its physical counterpart.

Figure 3 depicts an illustration of the setup in the 2D/Screen condition (a) and the 3D/VR condition (b). In both conditions, the participants sat on a chair in the laboratory, which was equipped with an EEG sensor net. In the 2D condition, a 27-inch monitor with a resolution of 1920 × 1080 pixels and a 60 Hz refresh rate was placed on the cupboard in front of the chair (Figure 3b). In contrast, the participants in the VR condition were equipped with a wired HTC Vive head-mounted display (HMD, 1080 × 1200 pixel resolution per eye) which was attached on top of the EEG sensor net (see Figure 2b/Figure 3b).

The same study prototype was used for both conditions and visualized the stimuli in the same virtual environment. For the 2D condition, an imaginary camera was activated at the level of the participant’s head to “film” the virtual environment. The “video footage” of the imaginary camera was displayed on the monitor screen positioned at the location where the stimuli were visualized in the virtual environment (i.e., on top of the TV cupboard). Thus, we could ensure that all visual parameters were the same in both conditions (e.g., background lighting and appearance of stimuli). The distance (110 cm), size (25 cm) and correspondingly also the visual angles (13°) of the stimuli, i.e., the faces, were controlled so that the subjects could perceive the faces of the avatars in the same way on the screen and in VR. In the VR condition, the participants “entered” the virtual environment immersively and the HMD controlled the position and orientation of their field of view. To reduce movement artefacts and ensure central presentation of the stimuli, participants were asked to keep their head still and center their focus on the center of the screen. If no face was present, a visual anchor was shown in the center of the place where otherwise the face would appear.

Importantly, while the actual stimuli were controlled, the visual angle of the surrounding area differed: The monitor had a height of 33 cm with a vertical visual angle of 17° and a width of 65 cm with a horizontal visual angle of 33°. The visual angle of the HTC Vive is much wider with 110° vertical and 100° horizontal, so participants were able to see more e.g., of the body of the avatars. Consequently, the resolution of the stimuli also differed, as the 1920 × 1080 pixel of the monitor are viewed in a smaller visual angle than the 1080 × 1200 pixel of the HTC Vive and thus, the stimuli had a higher resolution in 2D.

### 2.3. Procedure & Task

Each participant performed an eyesight test both in VR and 2D to ensure faces could be accurately perceived and the VR setup was correctly adjusted. The experiment consisted of two conditions: VR and 2D picture counting. In both conditions, participants viewed each of the 15 stimuli 20 times in random order, for a total of 300 trials per condition. Each stimulus was presented for 1000 ms with a variable inter-trial interval of 2500–3200 ms. Each condition was split into 10 blocks with 25–35 trials each. In each block, participants were asked to silently count either angry or neutral faces. They would receive an additional 0.5 € bonus for each correctly counted block and 0.25 € for each block within a ±1 margin of error as incentive. After each block, the experimenter would ask the number of counted targets and the target category changed. All participants viewed both VR and 2D conditions, with the order of conditions being balanced across participants. Before the actual experiment, each participant performed a test run to ensure that they could correctly identify the stimuli and understood the task.

After both VR and 2D stimulation, participants were asked a subset of questions from the igroup presence questionnaire (IPQ) [27] and the simulator sickness questionnaire (SSQ) [28]. From the IPQ (http://www.igroup.org/pq/ipq/download.php), items SP5, INV2, INV3, INV4, REAL1, REAL2, were selected and an added question “I had the feeling of sitting in front of a real person” with a 7 step scale from “not at all” to “very much”. From the SSQ, general discomfort, fatigue, boredom, drowsiness, eyestrain, difficulty focusing, and difficulty concentrating were selected with added questions for “how difficult was it to concentrate on the faces” and “how many trials do you estimate to have correctly counted”.

### 2.4. EEG Data Acquisition and Main Analysis

Brain and ocular scalp potential fields were measured with a 257 lead geodesic sensor net, on-line bandpass filtered from 0.01 to 100 Hz, and sampled at 1000 Hz using Netstation acquisition software and Electrical Geodesics (EGI, Eugene, OR, USA) amplifiers. Data were recorded continuously with the vertex sensor as the reference electrode. Stimulus synchronized epochs lasting from 1000 ms before until 2000 ms after picture onset were extracted. Heart and eye-blink artifacts were corrected by independent component analysis. Trials containing movement artifacts and noisy channels were rejected based on visual inspection of variance. For each participant, trials and channels with extreme variance were removed. On average, 8.2 (*SD* = 4.5) of the 256 channels were removed per participant. Removed EEG channels were interpolated by the mean of their neighboring sensors. The mean waveforms were calculated using on average 93% (*SD* = 2.6%) of the trials. After artifact correction, data were converted to an average reference. EEG data analysis was conducted using the open-source signal processing toolbox FieldTrip [29] and in-house functions using MATLAB 9.3.0 R2017b (MathWorks Inc., Natick, MA, USA).

Comparing signal quality between 2D and VR did not yield any significant differences (Table 1), neither did the sensor impedance differ, nor the number of excluded sensors or trials. Similarly, performance as measured by the number of correctly counted trials (±1 error) yielded no significant difference between 2D and VR.

#### 2.4.1. Frequency Analysis

Based on the results and methods from [21], EEG frequency analysis focused on induced brain activity. Accordingly, in a first step, the ERP average of each condition was computed and subtracted from the single trials comprising this condition. In a second step, a fast Fourier transform was calculated for single trial data in the frequency range from 1–40 Hz. Specifically, a sliding window of 200 ms was multiplied by a Hanning taper resulting in a spectral resolution of 5 Hz and a time resolution of ±100 ms. The sliding window advanced in 24 ms and 1 Hz increments to estimate changes in power over time and frequency. For each condition, single-trial power estimates were then averaged across trials. Values were expressed as a decibel change from pre-stimulus baseline (−300 ms to 0 ms) for stimulus locked analyses and as raw power for baseline comparisons.

#### 2.4.2. Statistics

To determine the main effects of emotion (angry vs. neutral) and task (target vs. non-target), data from each time point (0–2000 ms), sensor and frequency bin (1–40 Hz) of the respective picture/task condition were submitted separately to a dependent sample *t*-test. To account for the multiple comparisons problem, a cluster-based permutation test [30] was performed. In short, this procedure clusters adjacent *t*-values (in time, frequency and sensor space) to a single summed cluster test statistic. Clusters were formed when they had at least two neighbors which reached a cluster forming threshold of *p* < 0.05 (two-sided). These clusters were then tested against a Monte Carlo approximation of the test statistic, which was formed by randomly shuffling the data for 1000 permutations and reporting the proportion of random shuffles which were bigger as the actual observed cluster test statistic as a cluster *p*-value.

It is important to note that, as Sassenhagen and Draschkow [31] pointed out, the temporal, spatial and frequency boundaries of a cluster depend on the specific cluster-statistic settings and noise level in the data and as such should not be overestimated, e.g., across studies or different settings. To mitigate such concerns, effects are compared within the same study from the same preprocessed data with the same cluster settings.

Furthermore, exploratory follow up tests were conducted to reproduce the effects separately in each condition and test for interactions between conditions using the main effects of emotion and task as a region of interest. Towards this end, all sensors, time points and frequencies forming a significant cluster effect were summed up per condition and submitted to a Wilcoxon Signed-Rank Test to control for non-normal distributions.

#### 2.4.3. Event Related Potentials (ERPs)

ERPs were computed by averaging the raw data per condition and referring them to a −100 ms to 0 ms pre-stimulus baseline. Data were filtered with a highpass (−6 dB cutoff 0.5 Hz, transition width 1 Hz, order 828) and a lowpass filter (−6 dB cutoff 40 Hz, transition width 10 Hz, order 84). Both filters were zero-phase Kaiser windowed sinc FIR filters with a maximum passband deviation of 0.22% and a stopband attenuation of −53 dB (reporting as suggested by [32]). Afterwards, ERPs were subject to cluster analyses and tested within a 0–1000 ms time-window where the ERP effects were expected.

## 3. Results

### 3.1. Behavior and Questionnaires

As expected, facial expressions had a significant impact on both arousal and valence (Table 2). Post-hoc *t*-tests revealed that angry faces were rated as less valent and more arousing than neutral faces (valence *t*(15) = 12.5, *p* < 0.001; arousal: *t*(15) = 11.7, *p* < 0.001). Moreover, angry faces with an open mouth were also rated as more arousing and less valent than angry faces with a closed mouth (valence: *t*(15) = 11.4, *p* < 0.001; arousal: *t*(15) = 6.1, *p* < 0.001), which has been reported before [33]. Exploratory analysis indicated no gender differences of the emotional stimulus regarding valence and arousal ratings (*t*_s_ < 1.57, *p*_s_ > 0.16); however, addressing this issue conclusively requires larger sample sizes [34]. Accordingly, results focused on the strongest difference between neutral and angry faces with an open mouth.

When comparing valence and arousal ratings for avatars in 2D or VR, no significant differences were found. Comparing ratings of immersion and difficulty revealed a higher difficulty and more immersion for VR, as revealed by averaging over the questions of the igroup presence questionnaire (IPQ) for immersion and the simulator sickness questionnaire for difficulty (SSQ). The three questions with the biggest effect size of the IPQ were all related to the sense of being present and the (lack of) awareness of the real environment:INV3 “I still paid attention to the real environment.” (“Ich achtete noch auf die reale Umgebung”, *t*(15) = 4.56, *p* < 0.001, *d*_z_ = 1.14)INV2 “I was not aware of my real environment.” (“Meine reale Umgebung war mir nicht mehr bewusst”, *t*(15) = 3.12, *p* < 0.004, *d*_z_ = 0.78)SP5 “I felt present in the virtual space.” (“Ich fühlte mich im virtuellen Raum anwesend”, *t*(15) = 2.91, *p* = 0.005, *d*_z_ = 0.73).

The additional question about the “realness” of the actual stimulus was rated as marginally more significant: “I felt like sitting in front of a real person” (“Ich hatte das Gefühl, vor einer realen Person zu sitzen“) *t*(15) = 1.67, *p* = 0.057, *d*_z_ = 0.42. VR was also rated as more difficult than 2D overall, but not in all sub-questions. The effect was related to only two questions referring to blurry vision (“Haben Sie unscharf gesehen?” *t*(15) = 5.88, *p* < 0.001, *d*_z_ = 1.47) and difficulty focusing the eyes (“Hatten Sie Probleme Ihre Augen zu fokussieren?” *t*(15) = 2.18, *p* = 0.023, *d*_z_ = 0.55). No significant differences in general discomfort, fatigue, boredom, drowsiness, difficulty concentrating or difficulty of adhering to the task were observed.

Concerning task performance, both VR and 2D conditions had over 90% correctly counted trials with no difference between conditions, indicating an overall successful adherence to the task.

### 3.2. Induced EEG Frequencies

Comparing induced frequencies for angry and neutral faces revealed a significant (*p* = 0.012) main effect of emotion across widespread brain regions, centered around the alpha band (~8–15 Hz) between 1000–1800 ms (Figure 4). Even though the event related desynchronization between picture conditions started to differentiate already around 400 ms (Figure 4d), conventional cluster settings (*p* < 0.05) formed a significant cluster only after picture offset around 1000 ms. Noteworthy, exploratory tests within this cluster not only reproduced the effect separately for 2D (Angry *Mdn =* −0.68 dB, Neutral *Mdn =* −0.34 dB, *z* = −2.07, *p* = 0.039) and VR conditions (Angry Open *Mdn =* −0.84 dB, Neutral *Mdn =* −0.43 dB, *z =* −3.41, *p* < 0.001), but also revealed a significant interaction with stronger emotion effects in VR (2D Angry-Neutral Difference *Mdn =* −0.26 dB, VR Angry-Neutral Difference *Mdn =* −0.61 dB, *z =* 2.12, *p* = 0.034).

Furthermore, target pictures compared to non-target pictures revealed a significantly stronger (*p* = 0.025) alpha/lower beta-ERD (~10–20 Hz) for target pictures around 400–700 ms over central to central-left sensors (Figure 5). Similarly to the emotional ERD, exploratory post hoc tests reproduced the target main effect separately for 2D (Target *Mdn =* −0.86 dB, Nontarget *Mdn =* −0.66 dB, *z =* −2.43, *p* = 0.015) and VR conditions (Target *Mdn =* −0.76 dB, Nontarget *Mdn =* −0.26 dB, *z =* −3.52, *p* < 0.001) and also revealed an interaction with stronger target effects in VR (2D Target Difference *Mdn =* −0.19 dB, VR Target Difference *Mdn =* −0.49 dB, *z =* 2.17, *p* = 0.030).

Comparing VR vs. 2D induced frequencies directly revealed less beta-ERD (~16–28 Hz) for VR compared to 2D (*p* = 0.006), centered in posterior brain regions around 50–1200 ms (Figure 6). Moreover, the main effect was reproduced in exploratory post hoc tests separately for neutral (VR *Mdn =* −0.23 dB, 2D *Mdn =* −0.49 dB, *z =* 3.36, *p* < 0.001) and angry face conditions (VR *Mdn =* −0.17 dB, 2D *Mdn =* −0.55 dB, *z =* 3.15, *p* = 0.002) without significant interaction (*z =* −0.88, *p* = 0.379).

Additionally, 2D and VR conditions were also compared without baseline correction to estimate differences in raw frequency power in the baseline (Figure 7). For all higher frequencies (~10–40 Hz, Figure 7b), a consistent shift in frequency power over both pre- and post-stimulus time windows was observed (*p* < 0.001), with VR conditions exhibiting overall higher power but parallel time courses to the 2D frequencies (Figure 7d) over mainly posterior brain regions (Figure 7a). Exploratory post hoc tests reproduced the VR vs. 2D difference separately for neutral (2D *Mdn =* 0.65 µV^2^, VR *Mdn =* 1.03 µV^2^, *z =* −3.41, *p* < 0.001) and angry face conditions (2D *Mdn =* 0.62 µV^2^, VR *Mdn =* 1.04 µV^2^, *z =* −3.46, *p* < 0.001) without interactions between both (*z =* −0.52, *p* = 0.605).

### 3.3. ERPs

Combined across VR and 2D conditions, angry faces elicited a significantly increased (*p* = 0.021) late positive potential (LPP) over central regions, most apparent in a cluster around 300–500 ms (Figure 8). Exploratory post hoc tests (Figure 8c) reproduced this effect separately for VR (Angry *Mdn =* 1.27 µV, Neutral *Mdn =* 0.99 µV, *z =* 3.52, *p* < 0.001) and 2D (Angry *Mdn =* 1.02 µV, Neutral *Mdn =* 0.62 µV, *z =* 1.96, *p* = 0.049). Although the emotional difference in 2D (*Mdn =* 0.14 µV) was weaker than in VR (*Mdn =* 0.25 µV), the interaction was not significant (*z =* 1.09, *p* = 0.28).

In addition, comparing target to non-target pictures revealed a significant target P3 effect (*p* = 0.0049) over central regions around 400 to 650 ms (Figure 9). Exploratory post hoc tests (Figure 9c) reproduced this effect separately for VR (Target *Mdn =* 1.47 µV, Nontarget *Mdn =* 1.11 µV, *z =* 3.52, *p* < 0.001) and 2D (Target *Mdn =* 0.91 µV, Nontarget *Mdn =* 0.62 µV, *z =* 3.46, *p* < 0.001) with no significant interaction (*z =* 1.24, *p* = 0.21).

Directly comparing the ERP for VR vs. 2D stimulation revealed overall stronger ERPs in VR (Figure 10). When testing all sensors and timepoints while correcting for multiple comparisons via cluster analysis, a significant cluster (*p* = 0.049, Figure 10a) was found over posterior to central regions in an early time window around 100–200 ms. Exploratory post hoc tests reproduced this effect separately for neutral (VR *Mdn =* −0.11 µV, 2D *Mdn =* 0.74 µV, *z =* 3.46, *p* < 0.001) and angry faces (VR *Mdn =* −0.25 µV, 2D *Mdn =* 0.75 µV, *z =* 3.52, *p* < 0.001). Additionally, there was a significant interaction (*z =* 2.38, *p* = 0.017) with a stronger VR vs. 2D difference for angry faces (*Mdn =* −1.22 µV) compared to the difference for neutral faces (*Mdn =* −1.08 µV).

No additional clusters survived correction for multiple comparisons. However, based on the LPP and target P3 main effects in central regions around 300–650 ms, effects overlapping with these topographies and latencies were additionally considered. One cluster (*p* = 0.186, Figure 10b) overlapped with said LPP and P3 main effects in terms of latency (200–400 ms) and topography (posterior to central regions). In this cluster, VR also led to a stronger ERP in the P3/LPP time window. Moreover, exploratory post hoc tests reproduced this effect separately for neutral Neutral (VR *Mdn =* 2.10 µV, 2D *Mdn =* 1.36 µV, *z =* 3.52, *p* < 0.001) and angry faces (VR *Mdn =* 2.22 µV, 2D *Mdn =* 1.49 µV, *z =* 2.74, *p* = 0.006) with no significant interaction (*z =* 1.60, *p* = 0.108).

## 4. Discussion

To assess the effects of VR compared to conventional 2D presentations, the present pilot study considered several methodological aspects. First, by using angry and neutral facial expressions, the stimulus materials connect to previous 2D and VR studies [6,23,24,25]. Second, by including emotional and task manipulations, general and specific effects of VR stimulation can be assessed. Third, the analysis of brain oscillations and ERP components provides comprehensive insights into brain measures used in previous research to examine VR effects. Overall, our findings indicate that VR stimulation is associated with both, general and specific effects.

### 4.1. Research Question 1: General Effects Associated with VR Stimulation

The first goal of the present pilot study was to determine if virtual reality stimulation leads to response enhancement driven by increased immersion, or attenuated responding, reflecting distraction effects. Of note, for brain oscillations effects, VR stimulation was associated with an overall higher frequency activity both in pre- and post-stimulus periods, which leads to diminished event related desynchronization (ERD) relative to the baseline activity for all stimuli. This corresponds with earlier research showing dampening of distracting stimuli with higher immersion [9] or even using the powerful distraction provided by virtual reality as a means of dealing with pain [35].

For event related potentials (ERP), virtual reality led to overall stronger ERPs, both in an earlier time frame and in the time frame of the emotional LPP and target P3. This suggests greater cortical involvement in the processing of virtual reality stimulation.

Overall, the present findings provide an interesting case of virtual reality both enhancing and reducing effects, depending on the neural measurement. This underlines the importance of carefully considering multiple measurements and their implications, e.g., their baseline dependency: Event-related potentials isolate the time-locked or evoked components, which average out possible non-time locked events in the baseline. Contrary to ERPs, frequency analysis is also sensitive to non-time-locked or induced activity in the baseline. Depending on the research question, that might be beneficial or an unwanted contamination.

### 4.2. Research Question 2: Specific Effects of VR Stimulation on Emotion and Task Manipulations

The second goal of the present study was to assess whether VR stimulation has specific effects on emotion and task manipulations, i.e., does it specifically enhance or attenuate the processing of emotional and target stimuli?

For frequency effects, the attentional ERD modulation by both explicit tasks and implicit emotion was specifically enhanced by VR. This corresponds to the reported relationship between immersion and emotion [2,4] and suggests that virtual reality specifically enhances attentional capture by salient stimuli.

In ERPs, a greater difference between 2D and VR was observed for angry faces in an early time window, i.e., 100–200 ms. However, this interaction must be interpreted with caution, as no significant main effect for emotion was found in this early time frame. Given that absence of significance is not significance for absence, it could be that a sub-significance-threshold emotional effect was enhanced by VR. However, future research is needed to replicate and extend this finding.

Overall, both measurements showed a specific enhancement of emotional and task stimulus processing, which was most pronounced for frequency measures.

### 4.3. Control Analyses

Several control analyses replicated established emotion and attention effects. The main effects for emotion were conceptually replicated for valence and arousal ratings (the original study from Stolz et al. [6] measured threat which is similar to high arousal and low valence [36]). Moreover, emotional EEG effects were also replicated with angry faces eliciting a larger LPP [6,37] and more alpha-ERD [21]. Similarly, the explicit attention from a target mirrored the emotional effects, with a stronger P3 and alpha-ERD for target pictures. The replication of emotion and attention effects allowed for a meaningful interpretation of interaction effects with the stimulus modality.

Questionnaire data suggests that VR was rated as more immersive than 2D, which is seen as a key element in modulating emotion in VR [2,4]. Thus, the increased immersion in VR could be interpreted as the causal factor for the different EEG effects. However, VR was also perceived as blurrier and, surprisingly, 12 out of 16 subjects also reported a greater perceived distance to the stimulus in VR. Of note, a review indicated that distance judgement in VR is usually under- rather than overestimated [38]. Distance judgement is influenced by many factors, and in this case it might be that even though the faces were equally big and equally distant, VR allowed a bigger proportion of the rest of the body to be seen due to the increased visual angle in VR. Blurry vision and greater (perceived) distance would both normally predict smaller emotional effects [39]. Thus, immersion increasing emotional effects might even be partially occluded by the blurriness and perceived distance. Future studies should explore if equal blurriness and equal perceived distance could reveal the effects of immersion more clearly.

Apart from psychological differences, differences in the technical setup could also be potential factor for differences between VR and 2D. Sensor impedances for all sensors had a trend of being lower in VR possibly due to the added weight from the straps of the VR goggles; however, that trend vanished when only the included sensors (<100 kΩ impedance) were compared. Together with the equal number of included trials and sensors, differences in the signal to noise ratio are unlikely to have played a role in VR vs. 2D differences.

Another technical factor are the differences in displays: First, the HTC Vive had a lower latency between the analogue trigger and the actual stimulation picture, which was measured via a photo diode and corrected for in all analyses. Second, the 2D monitor had a 60 Hz refresh rate while the HTC Vive had a 90 Hz refresh rate with inserted black frames between normal frames. This black frame insertion should reduce motion blur and is especially used when the stimulation does not perfectly line up with the refresh rate [40]. These different modes of presentation might explain the morphological differences in the early ERPs (Figure 10d), i.e., that the ERPs in VR were not only stronger, but also steeper and partially earlier. However, even if the lower latency and black frame insertion might lead to more exact timings and thus different early potentials, such differences in the millisecond range cannot explain the differences in the later P3/LPP range (Figure 10h). In this later and longer time window, subtle timing differences should be averaged out. Thus, early P1/N1 ERP differences can possibly be attributed in part to timing/presentation differences, while P3/LPP effects are presumably unaffected.

Frequencies could also potentially be influenced by electrical noise from the head mounted display (HMD) in VR. However, the observed alpha/beta increase for VR vs. 2D was over occipital areas, where neither the VR goggles nor the straps to hold them were positioned. Rather, occipital areas are the main generators of (visual) alpha activity [41], so that the measured frequency differences are more likely to be generated by the brain rather than the technical setup.

A limitation of the present study is that it tested only few participants with basic stimuli and a reduced environment, that did not make use of much of the potential of VR (i.e., interacting with or moving in the virtual space). Nevertheless, to test 2D vs. VR stimulation, comparable stimuli are needed, and basic virtual environments reduce distractions. Even with this limited set of stimuli and participants, significant effects differentiating 2D and VR were found that could be even more enhanced with more sophisticated stimulations or more participants.

## 5. Conclusions

Taken together, the present study provides three main effects of VR vs. 2D stimulation: First, both ERP and frequency effects are modified in VR. While ERP components are overall stronger in VR, frequency results also show a different, more engaged baseline prior to stimulation. Second, frequency effects of implicit emotion and attention are enhanced in VR, suggesting that the increased immersion in VR also leads to stronger cortical engagement specific to salient stimuli. Third, VR stimulation is not better in every regard, but blurry vision, greater perceived distance and more frequency activity in the baseline probably due to distraction potentially diminished the effectiveness of VR. However, even with these detrimental effects, VR overall showed stronger effects of emotion and attention processes as compared to conventional 2D stimulation.

## Figures and Tables

**Figure 1 brainsci-10-00537-f001:**
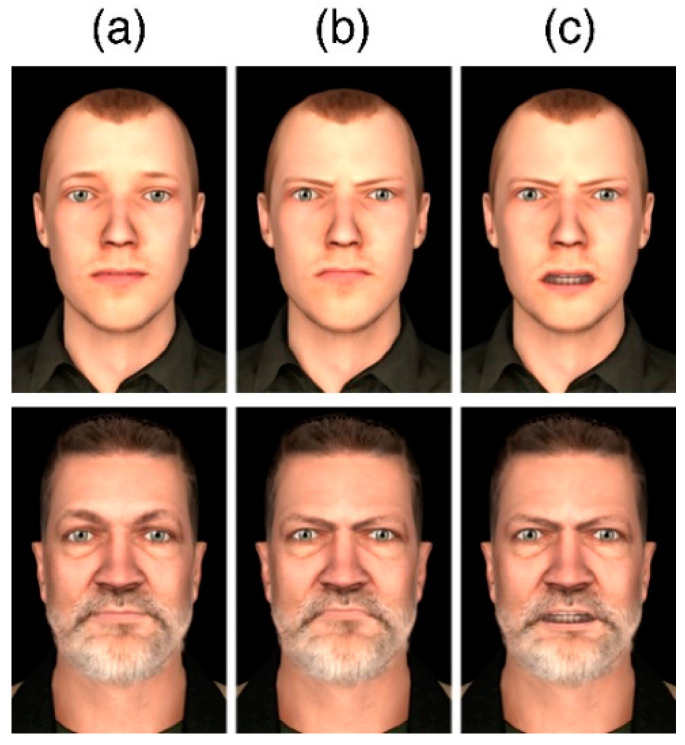
Two exemplary characters from the pool of avatars used in the study, each in all three conditions.

**Figure 2 brainsci-10-00537-f002:**
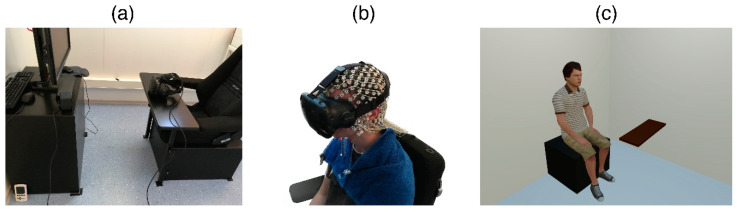
(**a**): laboratory where the experiment took place. (**b**): participant equipped with EEG net and VR head-mounted display (HMD). (**c**): virtual reconstruction of the laboratory with exemplary stimulus.

**Figure 3 brainsci-10-00537-f003:**
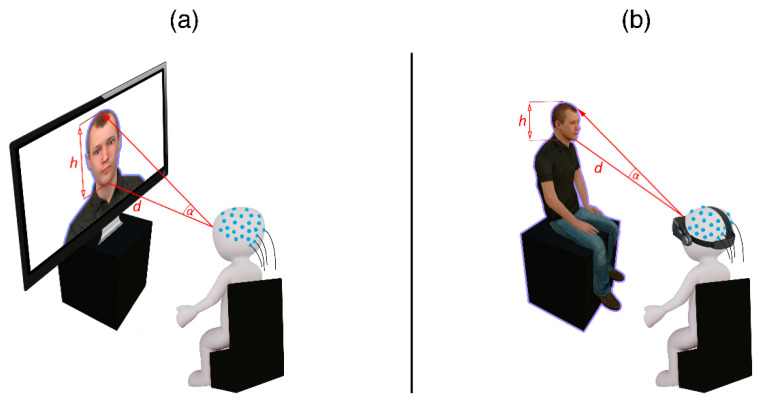
Illustrative setup of the two conditions 2D (**a**) and VR (**b**). The participants sat in a chair. Stimuli were displayed on a monitor screen (2D), or in Virtual Reality (VR). Stimulus size, distance and viewing angle for the face were controlled for in both conditions (h = 25 cm, d = 110 cm, α = 13°).

**Figure 4 brainsci-10-00537-f004:**
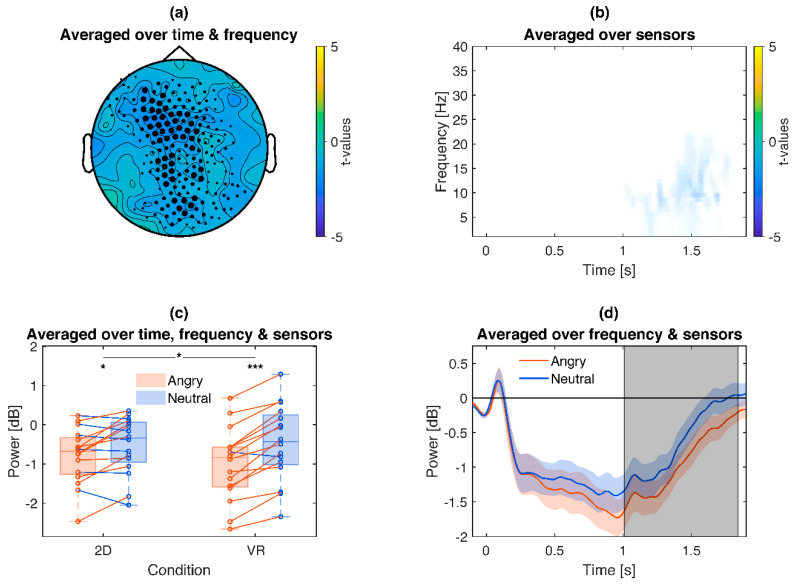
Illustration of the emotional frequency effects: angry pictures lead to a stronger alpha event related desynchronization (ERD), which is even stronger in VR. (**a**): Topography of *t*-values averaged across the significant time points and frequency bins (see b). The size of the marked sensors is displayed proportional to their contribution to the cluster. (**b**): Time-frequency plot of *t*-values, averaged across sensors of the respective cluster (see a). Only time-frequency bins which are part of the cluster are displayed. Opaqueness represents the percentage of sensors showing the effect, e.g., nearly opaque time-frequency bins indicate that only few sensors contribute to this effect. (**c**): Box plot and single subject values from the cluster. Red lines indicate a stronger ERD for angry compared to neutral pictures, blue the opposite. Asterisks indicate the significance of exploratory post-hoc group comparisons: *** *p* < 0.001, * *p* < 0.05. (**d**): Time course of the respective ERD cluster. Values were averaged over the respective sensors (see a) and frequencies (see b). Colored shaded areas signify standard errors. Grey shaded areas signify the latency of the cluster.

**Figure 5 brainsci-10-00537-f005:**
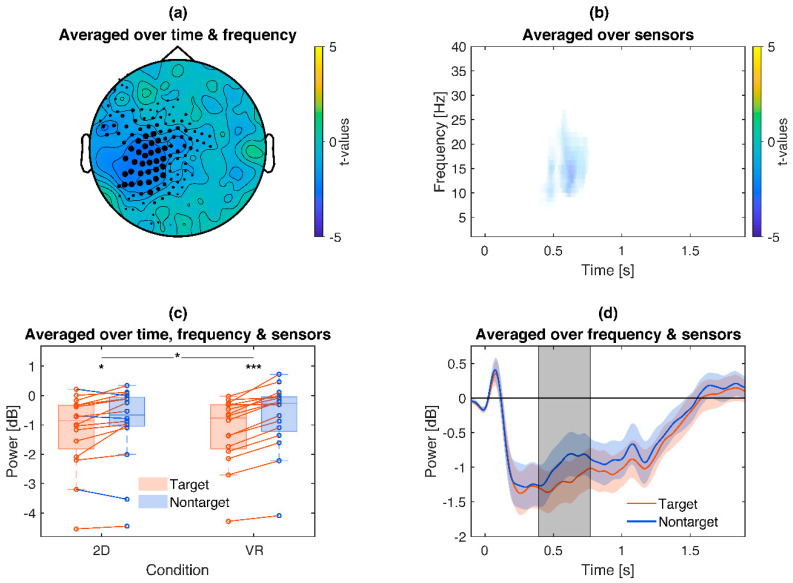
Illustration of the target frequency effects: Target pictures lead to a stronger alpha event related desynchronization (ERD), which is even stronger in VR. (**a**): Topography of t-values averaged across the significant time points and frequency bins. (**b**): Time-frequency plot of t-values, averaged across sensors of the respective cluster. Only time-frequency bins which are part of the cluster are displayed. (**c**): Box plot and single subject values from the cluster. Red lines indicate a stronger ERD for target compared to nontarget pictures, blue the opposite. Asterisks indicate the significance of exploratory post-hoc group comparisons: *** *p* < 0.001, * *p* < 0.05. (**d**): Time course of the cluster. Colored shaded areas signify standard errors. Grey shaded areas signify the latency of the cluster.

**Figure 6 brainsci-10-00537-f006:**
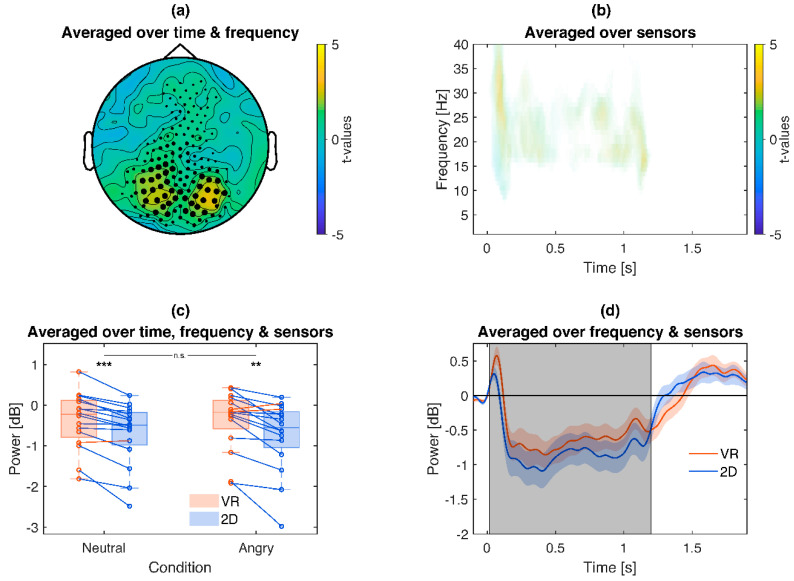
Illustration of the VR vs. 2D frequency cluster (baseline corrected): VR stimulation leads to overall weaker beta-ERD. (**a**): Topography of *t*-values averaged across the significant time points and frequency bins. (**b**): Time-frequency plot of *t*-values, averaged across sensors of the respective cluster. Only time-frequency bins which are part of the cluster are displayed. (**c**): Box plot and single subject values from the cluster. Blue lines indicate a weaker ERD for VR compared to 2D conditions, red the opposite. Asterisks indicate the significance of exploratory post-hoc group comparisons: *** *p* < 0.001, ** *p* < 0.01, ns *p* > 0.05. (**d**): Time course of the cluster. Colored shaded areas signify standard errors. Grey shaded areas signify the latency of the cluster.

**Figure 7 brainsci-10-00537-f007:**
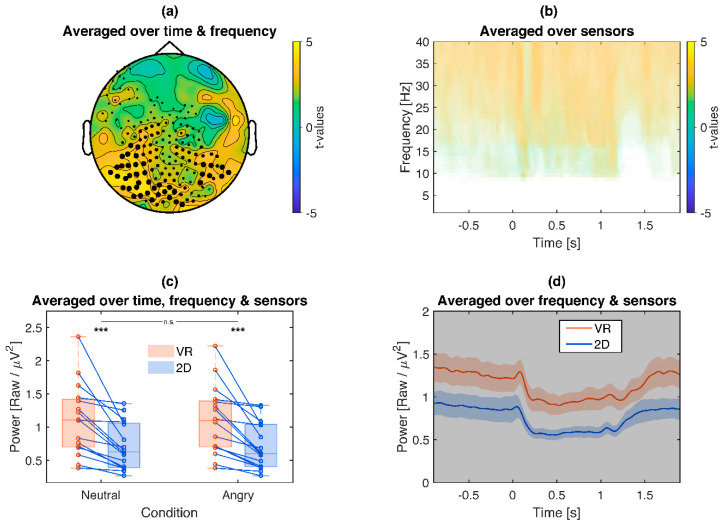
Illustration of the VR vs. 2D frequency cluster (raw power without baseline correction): VR leads to overall more high frequency power both in pre- and post-stimulus time periods. (**a**): Topography of *t*-values averaged across the significant time points and frequency bins. (**b**): Time-frequency plot of *t*-values, averaged across sensors of the respective cluster. Only time-frequency bins which are part of the cluster are displayed. (**c**): Box plot and single subject values from the cluster. Blue lines indicate more high frequency power for VR compared to 2D conditions, red the opposite. Asterisks indicate the significance of exploratory post-hoc group comparisons: *** *p* < 0.001, ns *p* > 0.05. (**d**): Time course of the cluster. Colored shaded areas signify standard errors. Grey shaded areas signify the latency of the cluster.

**Figure 8 brainsci-10-00537-f008:**
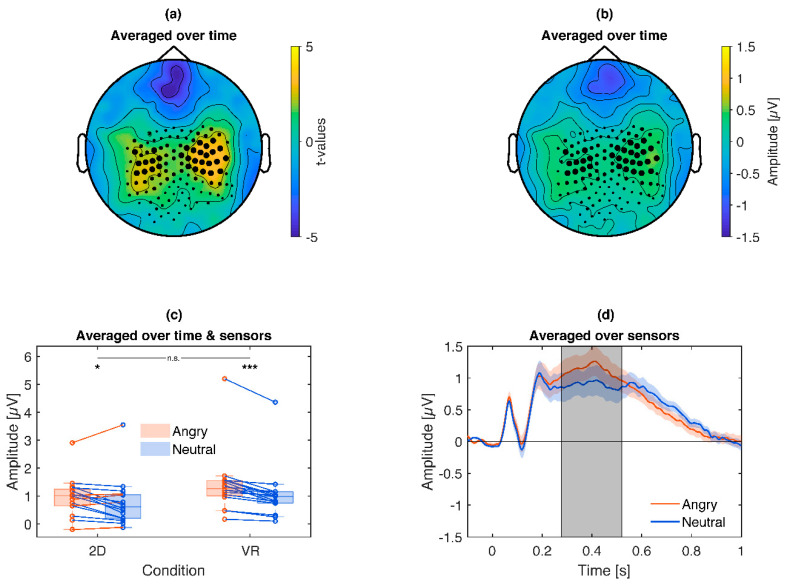
Illustration of the emotional modulation of the late positive potential (LPP) cluster: Angry pictures lead to an enhanced LPP compared to neutral pictures. (**a**): Topography of *t*-values averaged across the significant time points. The size of the marked sensors is displayed proportional to their contribution to the cluster. (**b**): Topography of the scalp potential difference. The size of the marked sensors is displayed proportional to their contribution to the cluster. (**c**): Box plot and single subject values from the cluster. Blue lines indicate a more positive LPP for angry compared to neutral pictures, red the opposite. Asterisks illustrate the significance of exploratory post-hoc subgroup comparisons: *** *p* < 0.001, * *p* < 0.05, ns *p* > 0.05. (**d**): Time course of the respective LPP cluster. Colored shaded areas signify standard errors. Grey shaded areas signify the latency of the cluster.

**Figure 9 brainsci-10-00537-f009:**
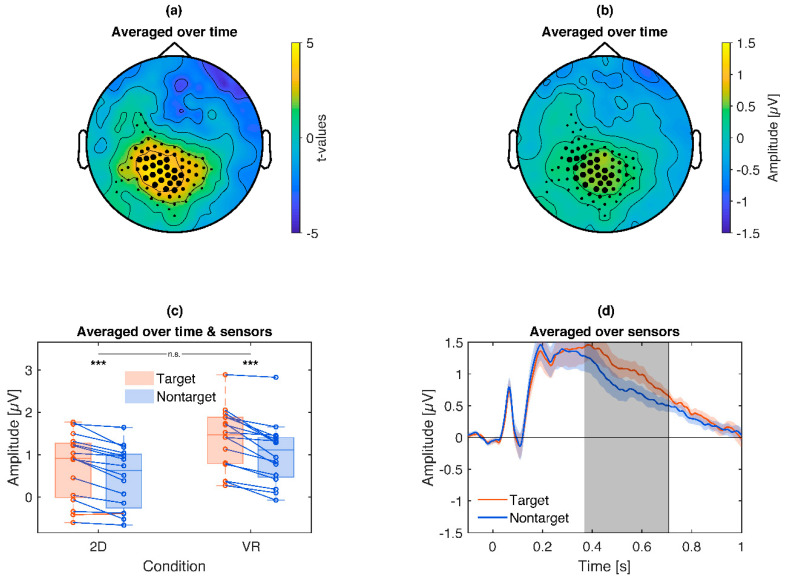
Illustration of the target P3 cluster: Target pictures lead to an enhanced P3 compared to nontarget pictures. (**a**): Topography of *t*-values averaged across the significant time points. (**b**): Topography of the scalp potential difference. (**c**): Box plot and single subject values from the cluster. Blue lines indicate a more positive P3 for target compared to nontarget pictures, red the opposite. Asterisks illustrate the significance of exploratory post-hoc subgroup comparisons: *** *p* < 0.001, ns *p* > 0.05. (**d**): Time course of the respective P3 cluster. Colored shaded areas signify standard errors. Grey shaded areas signify the latency of the cluster.

**Figure 10 brainsci-10-00537-f010:**
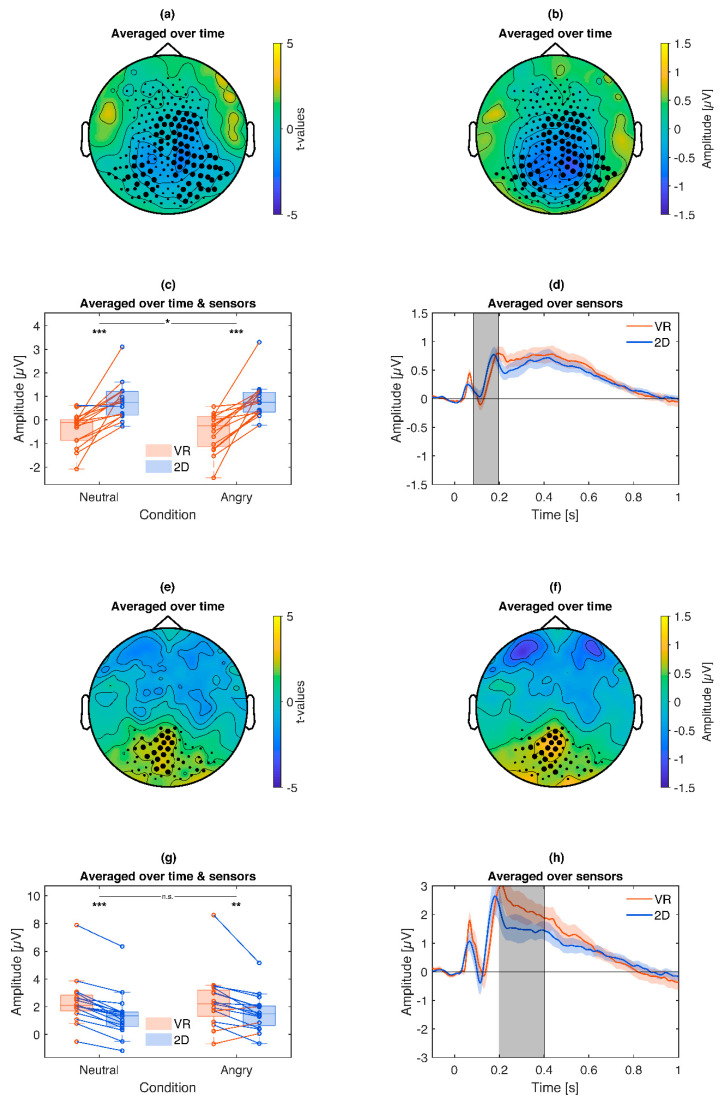
Illustration of the VR vs. 2D ERP effect: (**a**–**d**) VR stimulation leads to overall stronger ERPs and specifically more so for angry faces in an earlier time frame. (**e–h**) VR stimulation also leads to overall stronger ERPs in the time frame of the LPP and P3. (**a**) & (**e**): Topography of *t*-values averaged across the significant time points. (**b**) & (**f**): Topography of the scalp potential difference. (**c**) & (**g**): Box plot and single subject values from the respective cluster. Blue lines indicate a more positive ERP for VR compared to 2D conditions, red the opposite. Asterisks illustrate the significance of exploratory post-hoc subgroup comparisons: *** *p* < 0.001, ** *p* < 0.01, **p* < 0.05, ns *p* > 0.05. (**d**) & (**h**): Time course of the respective cluster. Colored shaded areas signify standard errors. Grey shaded areas signify the latency of the cluster.

**Table 1 brainsci-10-00537-t001:** EEG data quality.

Measurement	2D	VR	Statistical Comparison
Sensor impedance			
Overall	34.2 (236.3)	19.8 (119.8)	2D vs. VR *t*(15) = 1.40, *p* = 0.180, *d*_z_ = 0.35
Only < 100 kΩ	10.0 (6.1)	10.4 (6.6)	2D vs. VR *t*(15) = 0.99, *p* = 0.340, *d*_z_ = 0.25
Excluded sensors	7.8 (5.7)	8.6 (3.2)	2D vs. VR *t*(15) = 0.59, *p* = 0.561, *d*_z_= 0.14
Valid trials	278.5 (6.8)	279.8 (8.6)	2D vs. VR *t*(15) = 0.65, *p* = 0.528, *d*_z_= 0.16

**Table 2 brainsci-10-00537-t002:** Description: behavior and questionnaires.

Measurement	2D	VR	Statistical Comparison
Valence			2D vs. VR F (1, 14) = 0.09, *p* = 0.78, η² = 0.01 Facial expression F (2, 28) = 120.41, ***p* < 0.001**, η² = 0.90;
Neutral	3.45 (0.21)	3.56 (0.21)	
Angry Closed	1.91 (0.13)	1.95 (0.13)	
Angry Open	1.35 (0.95)	1.31 (0.95)	
Arousal			2D vs. VR F (1, 14) = 1.11, *p* = 0.31, η² = 0.07 Facial expression F (1.3, 17.5) = 127.72, ***p* < 0.001**, η² = 0.90;
Neutral	1.35 (0.21)	1.71 (0.21)	
Angry Closed	3.03 (0.29)	3.39 (0.29)	
Angry Open	3.91 (0.31)	4.24 (0.31)	
IPQ/Immersion	3.24 (1.08)	4.01 (0.89)	2D vs. VR *t*(15) = 2.48, ***p* = 0.026**, *d*_z_ = 0.62
SSQ/Difficulty	2.57 (0.50)	3.05 (1.65)	2D vs. VR *t*(15) = 3.33, ***p* = 0.005**, *d*_z_ = 0.83
Counted trials			
Correct ±1	94% (9%)	90% (14%)	2D vs. VR *t*(15) = 0.56, *p* = 0.292, *d*_z_ = 0.14

Note. Degrees of freedom have been adjusted via Greenhouse–Geisser correction for violations of sphericity. Bold font indicates significance at the *p* < .05 level. IPQ: igroup presence questionnaire. SSQ: simulator sickness questionnaire.

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
