# Peer review of "Virtual Reality Potentiates Emotion and Task Effects of Alpha/Beta Brain Oscillations"

_brainsci, 2020, doi:10.3390/brainsci10080537_

Round 1

Reviewer 1 Report

Indeed this is a very interesting article to read even during the summer holidays. This paper compares the VR and 2D stimuli from neural psychological perspectives.
The paper is easy to read and follow. I have though few recommendations for further improvement.
You have not mentioned why you have picked these two emotions in your measurements (Angry and neutral) for example why not fear? Second I was interested to know about your subjects' profile, Have there been any differences between ages, gender, and previous experience with VR.
I think the title of your paper is very long and not very descriptive to what you have done and presented in the paper.
A very long sentence at the end of the first paragraph in the Introduction, please revise.
I prefer that the introduction and related research should be separated into two different sections. Your elaboration of previous studies is very good, I recommend to create a summary table at the end of the introduction.
For more clarity and coherence add a new section Research Questions and Research method and move the research questions to this section. You may explain more about your subject's profile in this section.
Have both test participants were the same? does the test users know in advance what they are going to test? Do they have a perception of what they are expected to do?
Your result section is well presented, the main ambiguity is the difference in valence and arousal for different genders. This would have been another contribution of your paper.
I recommend in the discussion section add the research question and explain your finding based on the initially set questions

Author Response

We thank the reviewer for the helpful comments. We have revised the original draft accordingly and marked all such changes in red in the revised draft.

You have not mentioned why you have picked these two emotions in your measurements (Angry and neutral) for example why not fear?

Thank you for raising this issue which was very helpful to improve the manuscript. We picked these two emotions for two reasons: First, to connect the present study to previous research measuring ERPs to angry vs. neutral facial expressions. These studies reveal that angry faces evoke emotional electrocortical responses (e.g. [1,2,3]). Furthermore, there is evidence supporting that angry facial expressions are less prone to the uncanny valley effect compared to fearful facial expressions [4]. Second, in order to improve replicability we used stimuli that have been proven to evoke emotional responses in a previous Virtual Reality study with a setup comparable to our study [5]. In this study, angry facial expressions were thoroughly validated on self-report measures in an online pilot-study (and later in Virtual Reality). Thus, selecting these two emotions integrated previous research on conventional 2D and virtual reality stimulus presentation. We provide this information in the revised Method and Discussion section.

Second I was interested to know about your subjects' profile, Have there been any differences between ages, gender, and previous experience with VR.

In the revision, we provide the reader in the revision with descriptive statistics on age (between 19-31), gender (50/50 %) and previous VR experience (7 out of 16 participants had previous experience with virtual reality and within those, exposure to virtual reality in the last month than 1 hour in total). Small sample size and low variance prevented to systematically analyze whether and how differences in these variables impact VR stimulation effects.

I think the title of your paper is very long and not very descriptive to what you have done and presented in the paper.

We thank the reviewer for the suggestion and propose the following alternate title: “Virtual reality potentiates emotion and task effects of alpha/beta brain oscillations”

A very long sentence at the end of the first paragraph in the Introduction, please revise.

We thank the reviewer for bringing this issue to our attention and revised the sentence.

I prefer that the introduction and related research should be separated into two different sections. Your elaboration of previous studies is very good, I recommend to create a summary table at the end of the introduction. For more clarity and coherence add a new section Research Questions and Research method and move the research questions to this section. You may explain more about your subject's profile in this section.
Have both test participants were the same? does the test users know in advance what they are going to test? Do they have a perception of what they are expected to do?

We thank the reviewer for the suggestions to improve the manuscript. As suggested, we added a new section Research Question to direct the reader to the main aims of the present study. While a table on previous research would be informative, the complexity of the materials is difficult to comprehend in a summary table. Specifically, to capture heterogeneity among studies with respect to stimulus materials, methods, and brain measures would require a large or several tables, better suited to a review paper or meta-analyses. Furthermore, we improved the Method section. Specifically, we provide information on the subject’s profile. In addition, the reader is provided with the information that the participants were familiarized with the procedure before actual testing took part and clarified that we relied on a within-subject design.

Your result section is well presented, the main ambiguity is the difference in valence and arousal for different genders. This would have been another contribution of your paper.

We agree with the reviewer that gender differences are an interesting topic. However, this was not the main aim of this pilot study and would have required a substantially larger sample size. Acknowledging the limitation, we conducted exploratory analysis to test for gender differences on self-report measures. There were no differences in valence or arousal ratings between woman and men (ts < 1.57, ps > .16). Following your suggestion, we report the exploratory findings in the revised result section for completeness, indicating that addressing this issue conclusively requires larger sample sizes.

I recommend in the discussion section add the research question and explain your finding based on the initially set questions

We thank the reviewer for the suggestion and substantially revised the Discussion section explicitly focusing on the research questions.

  1. Duval, E.R.; Moser, J.S.; Huppert, J.D.; Simons, R.F. What’s in a face?: The late positive potential reflects the level of facial affect expression. J. Psychophysiol. 2013, 27, 27–38, doi:10.1027/0269-8803/a000083.
  2. Schupp, H.T.; Junghöfer, M.; Öhman, A.; Weike, A.I.; Stockburger, J.; Hamm, A.O. The facilitated processing of threatening faces: An ERP analysis. Emotion 2004, 4, 189–200, doi:10.1037/1528-3542.4.2.189.
  3. Smith, E.; Weinberg, A.; Moran, T.; Hajcak, G. Electrocortical responses to NIMSTIM facial expressions of emotion. Int. J. Psychophysiol. 2013, 88, 17–25, doi:10.1016/j.ijpsycho.2012.12.004.
  4. Tinwell, A.; Grimshaw, M.; Nabi, D.A.; Williams, A. Facial expression of emotion and perception of the Uncanny Valley in virtual characters. Comput. Human Behav. 2011, 27, 741–749, doi:10.1016/j.chb.2010.10.018.
  5. Stolz, C.; Endres, D.; Mueller, E.M. Threat-conditioned contexts modulate the late positive potential to faces—A mobile EEG/virtual reality study. Psychophysiology 2019, 56, 1–15, doi:10.1111/psyp.13308.

Reviewer 2 Report

The paper will be invaluable resource for the specialized audience. Hence, the paper is highly recommended for inclusion in the journal.

Author Response

We thank the reviewer for his positive review and recommendation of our paper.